# Revealing electronic state-switching at conical intersections in alkyl iodides by ultrafast XUV transient absorption spectroscopy

Kristina F. Chang [1], Maurizio Reduzzi[1], Han Wang [2], Sonia M. Poullain [1,3], Yuki Kobayashi [1], Lou Barreau[1,2], David Prendergast [2,4], Daniel M. Neumark [1,2] & Stephen R. Leone [1,2,5 ✉]

Conical intersections between electronic states often dictate the chemistry of photoexcited molecules. Recently developed sources of ultrashort extreme ultraviolet (XUV) pulses tuned to element-specific transitions in molecules allow for the unambiguous detection of electronic state-switching at a conical intersection. Here, the fragmentation of photoexcited *iso*-propyl iodide and *tert*-butyl iodide molecules ($i$-$C_3H_7I$ and $t$-$C_4H_9I$) through a conical intersection between $^3Q_0/^1Q_1$ spin–orbit states is revealed by ultrafast XUV transient absorption measuring iodine $4d$ core-to-valence transitions. The electronic state-sensitivity of the technique allows for a complete mapping of molecular dissociation from photoexcitation to photoproducts. In both molecules, the sub-100 fs transfer of a photoexcited wave packet from the $^3Q_0$ state into the $^1Q_1$ state at the conical intersection is captured. The results show how differences in the electronic state-switching of the wave packet in $i$-$C_3H_7I$ and $t$-$C_4H_9I$ directly lead to differences in the photoproduct branching ratio of the two systems.

[1] Department of Chemistry, University of California, Berkeley, CA 94720, USA. [2] Chemical Sciences Division, Lawrence Berkeley National Laboratory, Berkeley, CA 94720, USA. [3] Departamento de Química Física, Facultad de Ciencias Químicas,  Universidad Complutense de Madrid, 28040 Madrid, Spain. [4] Molecular Foundry, Lawrence Berkeley National Laboratory, Berkeley, CA 94720, USA. [5] Department of Physics, University of California, Berkeley, CA 94720, USA. ✉email: srl@berkeley.edu

The coupled evolution of electronic and nuclear structures plays a fundamental role in molecular reactions. The dynamics of chemical reactions are traditionally viewed within the Born–Oppenheimer approximation, which assumes separation of nuclear and electronic degrees of freedom in the system. While often applicable to reactions in ground electronic states, this approximation is frequently insufficient to describe excited state dynamics following photoexcitation where extensive couplings between electron and nuclear motions arise[1–4]. At crossings between the potential energy surfaces of electronic states where conical intersections are formed, the presence of strong couplings allow a molecule to abruptly transfer from one surface to another, thereby switching its electronic character. Due to the ubiquitous occurrence of crossings among electronically-excited states, the chemical outcomes of many photoinduced processes such as DNA photoprotection[5,6] and retinal isomerization in vision[7–9] are dictated by nonadiabatic state-switching at conical intersections.

Nonadiabatic dynamics at conical intersections are often challenging to capture experimentally as they necessarily involve multiple electronic states and typically evolve on a sub-picosecond timescale. Extreme ultraviolet (XUV) and soft X-ray absorption spectroscopies that measure resonant transitions from atomic core orbitals into unoccupied valence orbitals provide sensitivity to the symmetries, orbital occupations, and spin characteristics of electronic states[10–16]. Emerging femtosecond and attosecond transient absorption spectroscopies based on core-to-valence transitions therefore offer a powerful means of resolving multistate dynamics with excellent temporal resolution, enabling observations of rapid electronic state-switching at conical intersections which have previously eluded experimental observation.

The alkyl iodides (R-I, R = $C_nH_m$) constitute an important class of molecules for the investigation of nonadiabatic dynamics, as their dissociation in the $A$-band is intrinsically controlled by a conical intersection[17–24]. The $A$-band comprises dissociative spin–orbit states accessed by $5p \rightarrow \sigma^*$ valence excitation in the ultraviolet (UV) from a nonbonding iodine orbital into an antibonding orbital along the C–I bond. UV excitation results in rapid cleavage of the C–I bond within 200 fs[25]. Within the excitation, spin–orbit states carrying the Mulliken labels $^3Q_0$, $^1Q_1$, and $^3Q_1$ are optically accessible[17] (Supplementary Note 1). For few-carbon containing alkyl iodides, excitation to $^3Q_0$ comprises 70–80% of the oscillator strength in the $A$-band[21,26,27]. As shown schematically in Fig. 1a, UV excitation prepares an electronic–nuclear wave packet on the $^3Q_0$ surface correlating to the production of spin–orbit excited $I^*(^2P_{1/2})$ atoms. Throughout its motion along the steeply repulsive potential, a fraction of the initially prepared wave packet can cross to the $^1Q_1$ surface via a conical intersection, allowing for the release of ground state $I(^2P_{3/2})$ atoms. Consequently, the production of atomic I photoproducts has been primarily attributed to nonadiabatic $^3Q_0/^1Q_1$ state-switching via the conical intersection[19,21,28–30].

The I:I* photoproduct branching ratio varies widely among alkyl iodides depending on R-group structure. Previously-measured I:I* branching ratios obtained from the dissociation of several alkyl iodides at 277–280 nm are plotted in Fig. 1b. Methyl and ethyl iodide ($CH_3I$ and $C_2H_5I$) dissociation forms atomic I in a minority ratio of ~1:3 relative to I*[31,32]. In contrast, the dissociation of molecules with greater methyl substitution at the central carbon favor the release of atomic I. For $i$-$C_3H_7I$, atomic I photoproducts dominate in a ratio of ~2:1[33], while $t$-$C_4H_9I$ provides an even greater yield of ~13:1[29]. The dramatic increase in atomic I production suggests that a significantly larger fraction of the initial wave packet switches to the $^1Q_1$ surface while passing through the conical intersection[19,29,34]. While $i$-$C_3H_7I$ appears to

represent an intermediate case in which the wave packet bifurcates between the $^1Q_1$ and $^3Q_0$ states in a ~2:1 ratio, $t$-$C_4H_9I$ appears to represent a case of nearly-complete transfer to the $^1Q_1$ state. Owing to the importance of alkyl iodides as a benchmark system, direct observation of the conical intersection gating the formation of photoproducts is a very appealing target for both experiment and theory[35–37]. Although a number of time-resolved experiments on the dynamics of alkyl iodide photodissociation in the $A$-band have been reported using femtosecond XUV transient absorption and Coulomb explosion imaging[35,36,38,39], limitations in the temporal resolution of previously-reported experiments have precluded a direct observation of passage through the conical intersection.

In this report, ultrafast XUV transient absorption spectroscopy is applied to the investigation of the $^3Q_0/^1Q_1$ conical intersection dynamics in $i$-$C_3H_7I$ and $t$-$C_4H_9I$. Experimentally, dynamics are launched by a resonant femtosecond UV pump pulse and followed by a time-delayed attosecond XUV pulse that probes transitions between iodine I(4d) core orbitals and valence orbitals of the dissociating molecules (Fig. 1a). In the corresponding XUV absorption spectra, regions of the excited state surfaces both prior and subsequent to the conical intersection are mapped to distinct spectral features, which allows for the unambiguous detection of electronic state-switching at the conical intersection correlating with the release of atomic I* and I. The signatures of conical intersection dynamics and molecular fragmentation are found to be in excellent agreement with simulated XUV spectra of a $CH_3I$ model system.

## Results

**Time-resolved probing of *iso*-propyl and *tert*-butyl iodide.** The experimental pump-probe setup is summarized in Fig. 1c–e. Additional details of the experimental apparatus can be found in the "Methods" section. Briefly, gaseous $i$-$C_3H_7I$ and $t$-$C_4H_9I$ molecules in a quasi-static gas cell are excited by UV pump pulses (277 nm, 50 fs, 5 µJ per pulse) at a peak intensity of $1.1 \times 10^{12}$ W cm$^{-2}$. The UV pump spectrum is centered near the 260 and 268 nm $A$-band absorption maxima of $i$-$C_3H_7I$ and $t$-$C_4H_9I$, respectively[40]. Following UV excitation, dynamics are probed by time-delayed isolated attosecond XUV pulses (40–70 eV, ~170 as)[41] tuned to absorption transitions from the I(4d) core orbital appearing in the 45–48 eV photon energy range. A Gaussian instrument response function of 50 ± 7 fs (full width at half maximum) of the transient absorption experiment is measured using an in situ UV-XUV cross-correlation method.

As shown in Fig. 1a, wave packet motion from the highly repulsive region of the excited state surfaces (Regions 1–2) into the asymptotic dissociation limit (Region 3) is probed through XUV absorption transitions corresponding to core-to-valence excitations primarily localized on iodine. The $^3Q_0$ and $^1Q_1$ excited states are characterized by the configuration $(4d)^{10}...(\sigma)^2(5p)^3(\sigma^*)^1$ where the nonbonding $5p$ valence orbitals on iodine possess $5p\pi^*$ character due to interactions with the alkyl moiety. In a one-electron transition picture[35,38,39], the excited states can be probed by the excitation of available $4d \rightarrow 5p$ transitions to distinct $(4d_{3/2})^{-1}\sigma^*$ and $(4d_{5/2})^{-1}\sigma^*$ core-excited states separated in energy by the $4d$ core-hole spin–orbit splitting. In the corresponding XUV absorption spectra, transitions appear as doublets with excitations to $(4d_{3/2})^{-1}\sigma^*$ appearing at higher photon energies compared to excitations to $(4d_{5/2})^{-1}\sigma^*$. In Fig. 1a, stronger and weaker XUV transitions are distinguished by solid and dashed arrows. In this study, the $^3Q_0$ state primarily undergoes strong transitions to the $(4d_{3/2})^{-1}\sigma^*$ state appearing at higher XUV energies in the spectrum, whereas the $^1Q_1$ state primarily undergoes strong transitions to the $(4d_{5/2})^{-1}\sigma^*$ state

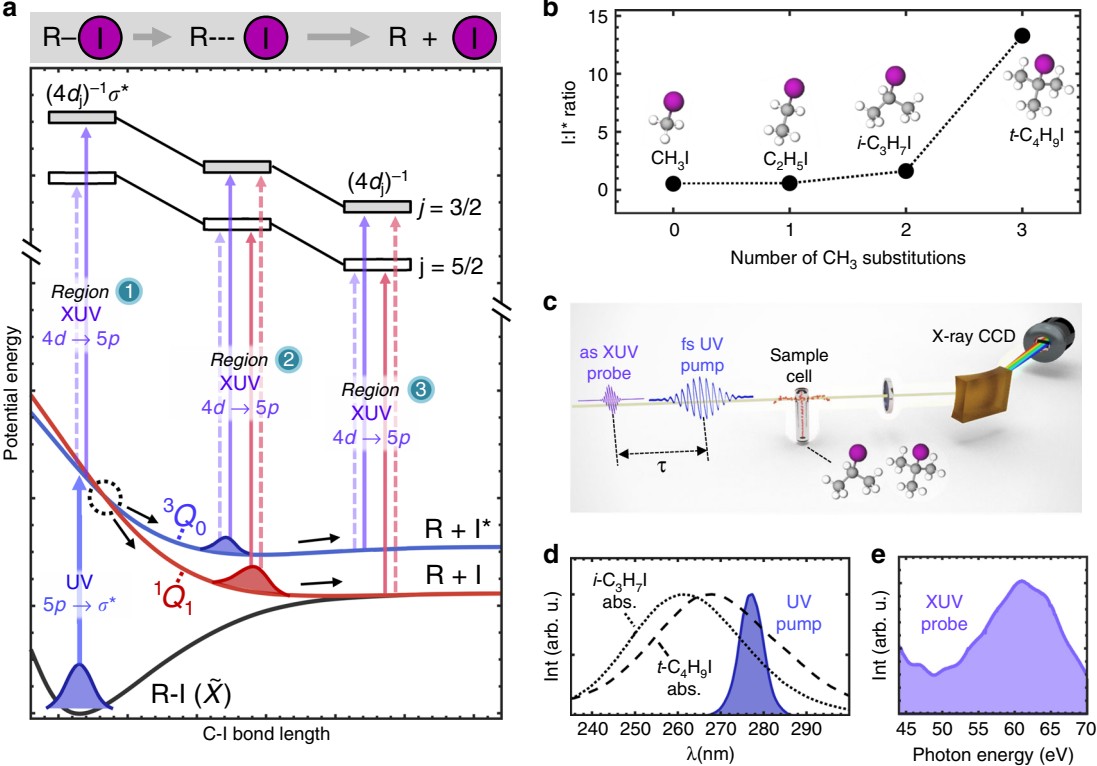

**Fig. 1 A-band fragmentation of alkyl iodides and experimental outline. a** Potential energy curves adapted from ref. [25] are plotted as a function of C–I distance. The $^3Q_0/^1Q_1$ conical intersection (dotted circle) allows for wave packet bifurcation into I* and I dissociation channels. The partitioning of available energy into different degrees of freedom is not represented in this schematic. Dynamics along the $^3Q_0$ and $^1Q_1$ potentials before the conical intersection (Region 1), after the conical intersection (Region 2), and in the dissociation limit (Region 3) are mapped through XUV transitions to core-excited molecular states labeled $(4d)^{-1}\sigma^*$. The molecular core-excited states connect to atomic $(4d)^{-1}$ core-excited states at large C–I distances and are plotted as repulsive based on their antibonding $\sigma^*$ character. **b** I:I* branching ratio data (277–280 nm excitation) obtained from refs. [29,31–33] and plotted as a function of methyl substitutions. The molecular structures of the alkyl iodides are depicted with carbon atoms in gray, hydrogen atoms in white, and iodine atoms in magenta. **c** Experimental UV-XUV pump-probe setup. After passing through the target sample, UV pulses are blocked by an aluminum filter and XUV pulses are transmitted, dispersed by a concave grating, and imaged onto an X-ray CCD camera. **d** UV pump spectrum and A-band absorption spectra of gaseous $i$-$C_3H_7I$ and $t$-$C_4H_9I$ adapted from ref. [40], and **e** XUV probe spectrum.

appearing at lower XUV energies due to spin–orbit selection rules imparted by the iodine atom. In addition to their primary appearance at distinct photon energies, dynamics along the $^3Q_0$ and $^1Q_1$ potentials are furthermore distinguished by their evolution at long-time delays. During molecular fragmentation along the C–I bond, the collapse of the hybridized molecular orbitals surrounding the iodine atom leading to a purely-atomic $(4d)^{10}...(5p)^5$ configuration is spectroscopically revealed through the convergence of molecular $^3Q_0$ and $^1Q_1$ features into peaks associated with free I* and I atoms at long-time delays.

Transient absorption spectra are recorded as changes in optical density $\Delta OD = -\log[I_{XUV+UV}(E, \tau)/I_{XUV}(E)]$, where $I_{XUV+UV}(E, \tau)$ is the XUV spectrum recorded at the time delay $\tau$ following the UV pump and $I_{XUV}(E)$ is the XUV spectrum recorded in the absence of the pump. The scan averages and integration times used to record the experimental transients are described in the "Methods" section. To eliminate high-frequency noise, the recorded transients are post processed using a low pass filter (Supplementary Note 2). After post processing, the experimental noise level is estimated as ~2 mOD. In Fig. 2a–d, the resulting transients for $i$-$C_3H_7I$ and $t$-$C_4H_9I$ are plotted between 44.5 and 48.5 eV photon energies where time-dependent features that reflect excited state dynamics are observed. Spectra plotted over the full photon energy range recorded (44–60 eV) can be found in Supplementary Figs. 1, 2.

Experimental spectra plotted at time delay intervals between −4 and 160 fs are shown in Fig. 2a, b. Several discrete, time-dependent features are observed in the spectra. The rich evolution of the features can be observed in the colormap depictions of the transient spectra shown in Fig. 2c, d. The convergence of features at early time delays (0–100 fs) into the fixed values of atomic transitions at longer times (100–160 fs) reflects dynamics evolving from the steeply repulsive to the asymptotic regions of the excited state potentials. In particular, dissociation in the asymptotic region (Region 3, Fig. 1a) is signified by the rise of well-known atomic transitions at 45.9 eV [I($^2P_{3/2} \rightarrow {}^2D_{5/2}$)], 46.7 eV [I*($^2P_{1/2} \rightarrow {}^2D_{3/2}$)], and 47.6 eV [I($^2P_{3/2} \rightarrow {}^2D_{3/2}$)][42,43]. In the $i$-$C_3H_7I$ transient, atomic I and I* peaks are clearly visible whereas in the $t$-$C_4H_9I$ transient, only the atomic I peaks are observed. The intensities of the observed atomic transitions allow I:I* branching ratio estimates of ~2:1 for $i$-$C_3H_7I$ and ≥9:1 for $t$-$C_4H_9I$ using the assumption that the I* signal is below the 2 mOD noise level of the experiment (Supplementary Note 3) and are consistent with the I:I* yields reported by previous measurements[29,33].

**Spectroscopic mapping of conical intersection dynamics.** Molecular features located at distinct photon energies from the atomic peaks in the XUV spectrum reveal dynamics in the repulsive region of the excited state potentials where the conical intersection is found (Regions 1–2, Fig. 1a). In both molecules

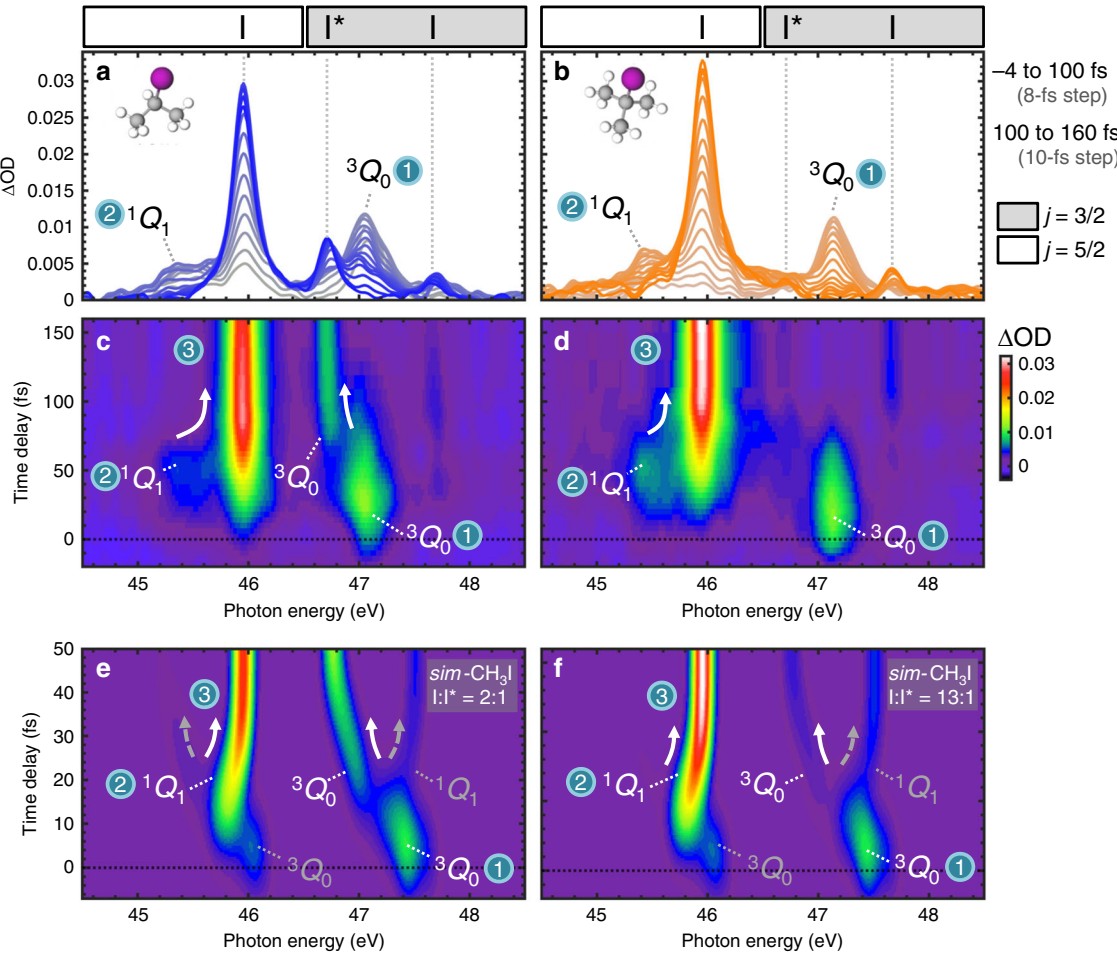

**Fig. 2 Experimental *i*-C$_3$H$_7$I and *t*-C$_4$H$_9$I transients and modified simulations of CH$_3$I transients.** Experimental spectra taken at selected time delays between −4 and 160 fs for **a** *i*-C$_3$H$_7$I and **b** *t*-C$_4$H$_9$I. The spectra are plotted in gray colors that evolve to blue or orange with increasing time delay. Dashed vertical lines indicate the positions of atomic iodine transitions. Regions of the spectra in which $(4d_{5/2})^{-1}$ and $(4d_{3/2})^{-1}$ core-level transitions appear are demarcated by white and gray boxes, respectively. Adjacent I(45.9 eV) and $^3Q_0$(47.1 eV) peaks overlap slightly in the 46.4–46.8 eV region. Experimental transients for **c** *i*-C$_3$H$_7$I and **d** *t*-C$_4$H$_9$I plotted as colormaps. State-specific molecular features and their convergence (indicated by arrows) to the atomic transitions are labeled according to the Region 1–3 labeling scheme introduced in Fig. 1a. Modified transient simulations of CH$_3$I converging to the empirical I:I* branching ratios of **e** *i*-C$_3$H$_7$I and **f** *t*-C$_4$H$_9$I. The simulations are temporally broadened by a Gaussian and independently normalized for comparison to the experiments.

(Fig. 2a–d), discrete molecular features located at 47.1 and 45.4 eV are transiently observed. While the two features both rise and decay within 100 fs, they otherwise exhibit different dynamics. The 47.1 eV feature is observed to maximize in intensity within the instrument response function. In contrast, the 45.4 eV feature maximizes in intensity 25 fs later, and its rise thus accompanies the decay of the 47.1 eV feature. Furthermore, the two features converge to different atomic absorption lines in the long-time limit. In the *i*-C$_3$H$_7$I transient where both I* and I peaks are observed, the 47.1 and 45.4 eV features exhibit clear shifts in energy into the I*(46.7 eV) and I(45.9 eV) lines, respectively (Supplementary Fig. 3). In the *t*-C$_4$H$_9$I transient where only I peaks are observed, the 45.4 eV feature shifts into the I(45.9 eV) absorption line whereas the 47.1 eV feature disappears with no accompanying I* rise. The molecular features at 47.1 and 45.4 eV are therefore assigned to $^3Q_0$ and $^1Q_1$ states, respectively. In accordance with its prompt appearance, the 47.1 eV ($^3Q_0$) feature is assigned to the region before the conical intersection (Region 1) directly populated by the UV pump. Meanwhile, the 45.4 eV ($^1Q_1$) feature is ascribed to the region after the conical intersection (Region 2) populated through nonadiabatic transitions from the $^3Q_0$ state at later times. Based on an analysis of

time traces taken at 47.1 eV in the molecular transients (Supplementary Fig. 4), $^3Q_0$ population is found to decay through state-switching at the conical intersection with a $\tau_{CI} = 36 \pm 4$ fs exponential time constant in *t*-C$_4$H$_9$I and $\tau_{CI} \leq 67 \pm 6$ fs exponential time constant in *i*-C$_3$H$_7$I (Supplementary Table 1). Although exponential time constants associated with state-switching at the conical intersection can also be obtained from time traces at 45.4 eV in the molecular transients, the analysis is currently limited by the temporal resolution of the experiments.

The concomitant decay and rise of the state-specific molecular signals in the XUV, appearing as a discontinuous switching of intensity from the $^3Q_0$ (Region 1) feature to the $^1Q_1$ (Region 2) feature in the transients, provides a clear spectroscopic signature of electronic reconfiguration at a conical intersection. Following passage through the conical intersection, dissociation dynamics along the $^3Q_0$ and $^1Q_1$ potentials (Regions 2–3) are mapped through continuous shifts into corresponding atomic I* and I absorption lines in the long-time limit. Thus, the ability of XUV spectroscopy to temporally and energetically resolve all electronic states involved in the reaction allows for the complete mapping of molecular fragmentation dynamics in the *A*-band, including the critical moment of wave packet bifurcation at the conical

intersection. The experimental XUV transients obtained exemplify the cases of intermediate and nearly-complete electronic reconfiguration at a conical intersection. In the case of $i$-$C_3H_7I$ (Fig. 2c), intermediate state-switching resulting in the partial retention of population on $^3Q_0$ after the conical intersection is directly observed as the spectral feature connecting the $^3Q_0$ (Region 1) signal to the atomic I* limit. In contrast, the nearly-complete transfer of population from $^3Q_0$ to $^1Q_1$ in $t$-$C_4H_9I$ (Fig. 2d) is signified by the abrupt disappearance of the $^3Q_0$ (Region 1) signal.

As shown in the experimental schematic in Fig. 1a, dynamics are followed through excitations to $(4d_{3/2})^{-1}\sigma^*$ and $(4d_{5/2})^{-1}\sigma^*$ core-excited states. The 47.1 eV ($^3Q_0$) signal can be assigned to $(4d_{3/2})^{-1}\sigma^*$ excitation based on its I*($^2P_{1/2} \rightarrow {}^2D_{3/2}$) convergence limit. Similarly, the 45.4 eV ($^1Q_1$) feature can be assigned to $(4d_{5/2})^{-1}\sigma^*$ excitation based on its I($^2P_{3/2} \rightarrow {}^2D_{5/2}$) convergence limit. In principle, complementary $^3Q_0$ and $^1Q_1$ transitions associated with $(4d_{5/2})^{-1}\sigma^*$ and $(4d_{3/2})^{-1}\sigma^*$ excitations, respectively, are also possible. Such transitions would appear as continuous signals connecting the 45.4 eV ($^1Q_1$) feature at early times to the atomically-forbidden 45.0 eV [I*($^2P_{1/2} \rightarrow {}^2D_{5/2}$)] convergence limit, and connecting the 47.1 eV ($^3Q_0$) feature to the weakly-allowed 47.6 eV [I($^2P_{3/2} \rightarrow {}^2D_{3/2}$)] convergence limit. However, no such signals are observed in the experimental results (Fig. 2a–d) and are presumed to be too weak to be detected. Instead, the selective probing of $^3Q_0$ transitions to the $(4d_{3/2})^{-1}\sigma^*$ core-excited state and $^1Q_1$ transitions to the $(4d_{5/2})^{-1}\sigma^*$ core-excited state results in the resolution of state-specific features at energetically distinct locations in the spectrum, thereby giving rise to a discontinuous appearance of electronic state-switching in the XUV spectra.

The assigned signatures of conical intersection and dissociation dynamics in $i$-$C_3H_7I$ and $t$-$C_4H_9I$ show strong resemblances to one another, as well as to those found in previously-simulated $CH_3I$ spectra. Similarities between the positions of the state-specific XUV features in the $i$-$C_3H_7I$ and $t$-$C_4H_9I$ spectra are consistent with expected similarities between their valence and core-excited states (Supplementary Fig. 5, Note 4) and motivate further comparisons to computed $CH_3I$ spectra. Simulated XUV transients representing the $A$-band dissociation of $CH_3I$ are directly obtained from ref. [37]. To facilitate comparisons to the experiments, the simulations are modified to reflect partial (I:I* = 2:1) and nearly-complete (I:I* = 13:1) electronic state changes upon passage through the conical intersection. In the modified $CH_3I$ transients (Fig. 2e, f), changes in the spectral features in photon energy and intensity show strong similarities to those observed in the $i$-$C_3H_7I$ and $t$-$C_4H_9I$ results. As in the experiments, bond-breaking dynamics are revealed through the convergence of molecular features into atomic limits. In the $CH_3I$ simulations, the convergence is completed more quickly as compared to $i$-$C_3H_7I$ and $t$-$C_4H_9I$, consistent with more rapid $CH_3I$ fragmentation[25]. The spectroscopic signature of $^3Q_0/^1Q_1$ conical intersection dynamics as a rise and decay of well-separated $^1Q_1$ and $^3Q_0$ features is also reproduced. As in the experiments, the selectivity of transitions from the $^3Q_0$ and $^1Q_1$ states to the $(4d_{3/2})^{-1}\sigma^*$ and $(4d_{5/2})^{-1}\sigma^*$ core-excited states, respectively, results in a characteristically discontinuous appearance of $^3Q_0/^1Q_1$ state-switching in the XUV spectrum.

## Discussion

Conical intersection dynamics in the alkyl iodides have long been a prototype for understanding nonadiabatic processes in photochemistry. In this work, nonadiabatic fragmentation dynamics of $i$-$C_3H_7I$ and $t$-$C_4H_9I$ are revealed by ultrafast XUV transient absorption spectroscopy. In both molecules, spectroscopic measurements from the perspective of core-to-valence excitations localized on iodine allow for a complete mapping of the chemical reaction from UV photoexcitation to photoproduct formation. The sensitive detection of transient molecular and atomic electronic states involved in the fragmentation pathway provides an exacting picture of ultrafast wave packet bifurcation between electronic states at a conical intersection. Specifically, XUV signatures portraying the cases of partial wave packet transfer in $i$-$C_3H_7I$ leading to an intermediate I:I* branching ratio and nearly-complete wave packet transfer in $t$-$C_4H_9I$ leading to the dominant formation of I atoms are captured. Furthermore, by comparisons to calculated spectra of a $CH_3I$ model system, the XUV signatures are shown to be readily interpretable within a straightforward, one-electron picture of core-to-valence transitions.

The present study demonstrates the general advantages of resonant photoexcitation combined with a direct probing of valence electronic structure in the XUV for capturing non-adiabatic electronic state-switching in polyatomic systems. Future experiments with shorter pump pulses, achieving faster temporal resolution, will allow time constants for the passage of $i$-$C_3H_7I$ through the conical intersection to be more precisely characterized and compared to $t$-$C_4H_9I$, providing further insight into the influence of alkyl group structure on state-switching dynamics. In addition, complementary experiments probing core-level transitions at the carbon K-edge[10,11,44] could allow for the detection of structural dynamics within the R-group moiety of the alkyl iodides during C–I dissociation, thus providing a multi-dimensional picture of passage through the conical intersection. Finally, the application of ultrafast XUV transient absorption methodologies to classes of molecules beyond alkyl iodides will continue to provide a powerful route for the direct investigation of non-Born–Oppenheimer dynamics governing the chemistry of electronically-excited systems.

## Methods

**Experimental setup**. The $i$-$C_3H_7I$ and $t$-$C_4H_9I$ molecules are obtained from Sigma-Aldrich at 99% and 95% purity, respectively. The sample target consists of a 3 mm long quasi-static gas cell filled to a pressure of ~5 Torr at room temperature (298 K). Alkyl iodide molecules in the gas phase are excited by UV pump pulses and probed by time-delayed attosecond XUV pulses.

Attosecond XUV probe pulses are generated by a table-top high harmonic setup[41]. The setup employs the output of a carrier–envelope phase stable Ti:Sapphire amplifier delivering 27 fs, near-infrared (NIR) pulses at a 1 kHz repetition rate. Spectral broadening of the pulses in a hollow-core fiber filled with neon, and subsequent compression by a combination of chirped mirrors and passage through an ammonium dihydrogen phosphate crystal and fused silica produces few-cycle, sub-4 fs pulses with a broadband spectrum extending from 500 to 900 nm. By focusing the few-cycle NIR pulses into a quasi-static gas cell filled with argon, isolated attosecond XUV pulses are generated through amplitude gating. The spectrum of the attosecond XUV pulses exhibits a smooth continuum structure between 40 and 70 eV. According to previous streaking measurements, the XUV pulse duration is estimated to be ~170 as[41]. Residual NIR light is subsequently removed from the XUV beam path by a 200 nm thick aluminum filter. The XUV pulses are then focused into the sample gas cell by a toroidal mirror, and the transmitted spectrum is dispersed by a concave grating and measured by an X-ray CCD camera. In this study, the photon energy range between 44 and 60 eV is mainly employed. The photon energies of the XUV spectrum are calibrated using well-known Fano resonances of neon between 40 and 50 eV[45]. By fitting the resonances to a Fano lineshape convolved with a Gaussian function representing the experimental spectral resolution, a spectral resolution of 40 meV (full width at half maximum) is estimated.

Femtosecond UV pump pulses are generated by sum-frequency mixing[46] between broadband and narrowband input pulses. The spectra of the input pulses and the configuration of the sum-frequency mixing setup are outlined in Supplementary Fig. 6 and Supplementary Note 5, respectively. The UV pump arm is loosely-focused to a spot size of 90 μm in the sample gas cell at a crossing angle of 0.7° with respect to the XUV arm. The UV beam after the sample cell is blocked before the X-ray CCD camera by a 200 nm thick aluminum filter. Time overlap of the UV pump and XUV probe pulses as well as the UV-XUV instrument response function are characterized in situ via the measurement of ponderomotive shifts in core-excited atomic xenon with the UV pulse modeled as a Gaussian[35,38].

Following this methodology, an instrument response function of 50 ± 7 fs is determined. Based on this, the UV pump pulses are anticipated to be ~50 fs in duration at the gas target.

Each time-dependent XUV spectrum of the $i$-$C_3H_7I$ experiment (Fig. 2a, c) and $t$-$C_4H_9I$ experiment (Fig. 2b, d) is obtained from an average of 70 and 50 X-ray camera frames, respectively. Each frame is captured at an integration time of 1 second per frame, 1000 laser pulses per second. In both experiments, XUV spectra are collected at time delays from −50 to 160 fs. Between −20 and 100 fs time delays, spectra are recorded at 4 fs intervals. Outside of this delay window (i.e., −50 to −20 fs and 100 to 160 fs), spectra are recorded at 10 fs intervals. The average standard deviation in ΔOD across the 44–60 eV photon energy range of interest in the XUV spectrum is $\sigma_{avg} = 2$ mOD and is interpreted as the noise level of the experiments.

**Simulation details.** The simulations used to produce theoretical $CH_3I$ transients for comparison to the experimental $i$-$C_3H_7I$ and $t$-$C_4H_9I$ results are published[37]. In the simulations, nonadiabatic dynamics of $CH_3I$ after photoexcitation to the $^3Q_0$ state were computed using Tully's fewest-switches surface hopping theory implemented in the SHARC software package. The resulting molecular dynamics trajectory data were used for the computation of XUV transient absorption spectra simulated with OpenMolcas using the MS-CASPT2 method and ANO-RCC-VTZP basis set. Molecular trajectories leading to the dissociation of I* and I atoms provide distinct signatures in the XUV transients, which are plotted in Supplementary Fig. 7. The modified $CH_3I$ transients shown in Fig. 2e, f are produced from direct sums of the XUV transients associated with I* and I dissociation, and with a Gaussian time-broadening applied (Supplementary Fig. 8, Note 6).

## Data availability

The data supporting the findings of this study are available from the corresponding author upon reasonable request.

## Code availability

The codes used to simulate the modified $CH_3I$ transients and analyze the experimental results are available from the corresponding author upon reasonable request.

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

## Acknowledgements

This work is supported by the National Science Foundation (nos. CHE-1361226 and CHE-1660417) (K.F.C., M.R., Y.K., and S.R.L.), the U.S. Army Research Office (no. W911NF-14-1-0383) (K.F.C., Y.K., D.M.N., and S.R.L.), and the U.S. Department of Energy under the Gas Phase Chemical Physics Program (no. DE-AC02-05CH11231) (H.W., L.B., D.P., and S.R.L.). Supercomputer time was provided by the National Energy Research Scientific Computing Center. S.M.P. acknowledges funding from the European Union's Horizon 2020 research and innovation program under the Marie Sklodowska-Curie grant agreement (no. 842539, ATTO-CONTROL). Y.K. acknowledges financial support by the Funai Overseas Scholarship. L.B. acknowledges the support of a Fellowship from the Miller Institute for Basic Research in Science, University of California Berkeley. We also thank A. Zanchet and A.R. Attar for useful scientific discussions, as well as C. Manzoni and R. Borrego-Varillas for discussions regarding the construction of the pump pulse setup.

## Author contributions

K.F.C., D.M.N., and S.R.L. conceived the experiments. K.F.C. performed the experimental measurements and analyzed the results. M.R. and K.F.C. constructed the experimental pump pulse setup. H.W. and D.P. performed the theoretical calculations. K.F.C., D.M.N., and S.R.L. wrote the paper with inputs from M.R., H.W., S.M.P., Y.K., L.B., and D.P.

## Competing Interests

The authors declare no competing interests.
