## [Peer Review File · Nature Communications]

REVIEWER COMMENTS

Reviewer #1 (Remarks to the Author):

This manuscript reports what is stated in the title, for two specific alkyl iodides – i-propyl iodide and t-butyl iodide. I am torn in my assessment. The work is noteworthy in that it provides (further) illustration of the insights into the photodissociation dynamics of small gas phase molecules that can be achieved using cutting-edge time resolved pump-probe spectroscopy methods (here using XUV absorption as the probe). Against this, I perceive two rival viewpoints. First, the opportunities afforded by this type of experiment have already been demonstrated, quite spectacularly in some cases, in other recent publications from the Berkeley group – which include papers describing studies of the simplest alkyl iodide (methyl iodide, ref. 36) and a somewhat richer variant, allyl iodide (ref. 40). Second, as the authors note in their introduction, the UV photochemistry of the alkyl (and related) iodides has already been much studied. What new insights are revealed in this manuscript? The key features, e.g. excited state potentials, relative transition dipole strengths, parent molecule excited state to atomic product state correlations, are all understood reasonably well and I/I* product branching ratios have been measured (often at several wavelengths) and rationalised in terms of branching at a conical intersection between the 3Q0 and 1Q1 potential energy surfaces – as summarised in the Introduction. Real time observation of the bifurcation at this conical intersection would be noteworthy but, as the authors point out (p. 3), ‘The determination of time constants characterising the evolution of the 3Q0 and 1Q1 features is currently limited by the temporal resolution of the experiments and is therefore not provided here’. Hence my dilemma: Those familiar with the field might well enjoy reading another well-written description of an application of an impressive new technology but will find few surprises in the results, while the more general chemical physics oriented reader could have already gained a similar overview from reading prior works from this group.

There is no doubt the science merits publication; my uncertainty is whether this particular work meets the stated criterion for publication in Nature Communications – namely that ‘it represents an advance in understanding likely to influence thinking in the field.’ But, given my expectation that the work will be published in some appropriate journal, I offer the following comments that the authors may wish to consider:

p. 2. The statement ‘For few-carbon containing alkyl iodides, excitation to 3Q0 comprises 70-80% of the oscillator strength of the A band’ may need qualifying. For comparison with the present data, one is interested in the partial absorption cross-sections to the various excited states at the pump wavelength of interest. If transitions to excited states other than 3Q0 in the present study really were responsible for 20-30% of the total oscillator strength, should they not be evident in the zero time delay transients?

Fig. 2. The authors might reflect on the colour scheme used to display in panels (a) and (b) (pale to dark with increasing time delay). This choice ensures that the dominant features are associated with the asymptotic I and I* atoms, yet the point of primary novelty is surely the variation in the transient

spectra at short time delays – which is currently hard to make out. Might a multi-colour scheme allow clearer visualisation of the early time data?

p. 5 contains two curious sentences: 'The modified CH₃I transients qualitatively reproduce with excellent agreement'; and 'Future experiments with even shorter pump pulses will be the subject of a future study.'

p. 9. D.M.N. initials sequenced wrongly in line 3 of Author Contributions.

Reviewer #2 (Remarks to the Author):

This paper presents a very nice time-resolved study of the photodissociation of isopropyl- and t-butyl-iodide into an alkyl radical and an iodine atom in the I or I* spin-orbit state. Ultrafast transient absorption spectroscopy is used to observe passage through a conical intersection along the dissociation coordinate. The switch between potential surfaces leads to a switch in the absorption spectrum of the dissociating molecule as well as that of the ultimate fragments. What is great about the paper is that it is very clearly written, the effects are clearly visible and (mostly) readily understood by the reader, the theory presents some confirmation of the model, and the whole thing could be used as a terrific textbook example of the effects of conical intersections on photodissociation dynamics. The demonstration of the experimental approach, especially when combined with the theory, clearly suggests that it will have substantial long term value, and I am sure it will lead to lots of exciting avenues of research. I am not sure one could point to any one thing in the results on alkyl iodide photodissociation where you could say, these results demonstrate something that nobody knew before, but the example is so nice and the technique is so powerful that I believe that the paper should be accepted for publication in Nature Communications. As I said, the paper is very clearly written, and could be published as is. I just have a couple points the authors might want to consider.

1. On p. 3 the authors say, "the 3Q0 state is observed primarily undergo strong transitions to the (4d_{3/2})-1σ* state..., whereas the 1Q1 state primarily undergoes strong transitions to the (4d_{5/2})-1σ* state..." This makes it sound like the selectivity just happens to be there, but it seems like there must be a simple atomic picture that shows why this is true (and certainly the calculations reproduce it). Maybe they could put something in the supplementary material about this, and then change the text to make it sound like they know why it is that way?

2. The pictures used in the paper show the motion along the reaction coordinate, but of course, even in the photodissociation of methyl iodide, the motion along other normal modes affects the

branching and product distributions (e.g., the calculations of Guo and Schatz). The authors mention in the discussion that probing near the carbon edge could provide insight into those dynamics, which would be really neat. It would be interesting to understand the extent to which such motions responsible for the changing branching ratios in different alkyl iodides. Certainly, the translational energy distributions for both samples show broad distributions (see Refs. 29 and 30), reflecting significant excitation of the radical vibrational modes. In any case, in the presence of all these other degrees of nuclear motion, it is interesting that the passage through the conical intersection still happens so fast.

3. I probably missed something in the explanation, but in Figure 2c, the population in 1Q1 evolves smoothly into the feature associated the free I atoms, but for the feature associated with 3Q0, there is a minimum of density in the evolution from the 3Q0 feature to the free I* atoms. This is reproduced in the simulation, but I am not quite sure why there is this minimum. Does it just reflect the very short period of time at the corresponding geometry? Why is it absent, or at least much less obvious, for the 1Q1 component?

These points are more curiosities, and I do think the paper can be accepted in the present form.

Reviewer #3 (Remarks to the Author):

The group is one of the pionner scientists to have developed the ATAS technique to reveal molecular dynamics. They have in the past studied several compounds using ATAS among which are some halogens :

- Direct Observation of the Transition-State Region in the Photodissociation of CH3I by Femtosecond Extreme Ultraviolet Transient Absorption Spectroscopy ». JPCL 2015 . (ref36)
- Transition State Region in the A-Band Photodissociation of Allyl Iodide—A Femtosecond Extreme Ultraviolet Transient Absorption Study ». JCP 2016 (ref40)
- Tracking Dissociation Dynamics of Strong-Field Ionized 1,2-Dibromoethane with Femtosecond XUV Transient Absorption Spectroscopy . PCCP(2016)
- Disentangling Conical Intersection and Coherent Molecular Dynamics in Methyl Bromide with Attosecond Transient Absorption Spectroscopy ». Nature Communications (2019) (ref15)

The group now presents a study of the iso-propyl iodide and tert-butyl iodide molecules (i-C₃H₇I and t-C₄H₉I) dissociation which is characterized by a larger amount of I/I* branching ratio relative to smaller alkyl iodides. The introduction, experimental results and their assignments are perfectly written and explained with the proper references. The reading is very easy and the experimental data beautiful to see (only 2mOD ! Thanks to the high-frequency post-process analysis). It was a real pleasure to read it. If the below mentioned concerns and minor items can be addressed satisfactorily, one could consider publication in Nat. Com.

Main concerns :

- 1) In Fig2c for i-C₃H₇I, why does the feature at 47.6 eV, that is fragment signature , seems dominant before 120fs and not after ?
- 2) In fig2d for t-C₄H₉I , what is the assignment of transients at 46.4-46.8 eV/40-90 fs ?
- 3) It is really disappointing that there is no attempt to fit the rising and decay times of the features by a deconvolution with the apparatus function. Indeed, when the precision of the cross-correlation is +/-7fs, it means that the resolution of the dynamics will appear on the same range. It is especially true to go beyond the contour plot that gives the misleading first impression that product features at 45.9 eV appears as soon as 0fs for i-C₃H₇I. Similar treatments were already made in the JCP 2016 for allyl iodine with a longer cross-correlation. If it is due to the limited space of Nat.Com, it could be included in the SI. It is also relevant to compare the two molecular systems and to conclude if the time-scales can also rationalize the different I*/I ratio observed.
- 4) I don't understand what is clearly done in 'enforcing the convergence of I:I* branching ratios to values of 2:1 and 13:1 at 50 fs time delay'. A few sentences should be written to tell the reader that the dynamics in the simulation is twice as fast as in the experiment. Could this simulation be temporally broadened by a 50 fs Gaussian (as in the experiment) instead of 15 fs which does not make sense to me.

Minor ones

- 5) I would have expected a slight energy variation of the 1 and 2 features due to the chemical shift variation with the C-I distance. This is not observed : is it due to the long 50fs duration of the pump pulse ? It would be great to discuss the requirement on the experimental set-up to observe on such

compounds a chemical shift variation.

6) I don't like the figure 1A because it give the wrong impression that R products appear without internal energy and that all the relased energy is of kinetic nature relative to the center of mass.

Normal text: reviewer comments

Orange text: our response

Blue text: revisions made

Response to Reviewer #1

This manuscript reports what is stated in the title, for two specific alkyl iodides – i-propyl iodide and t-butyl iodide. I am torn in my assessment. The work is noteworthy in that it provides (further) illustration of the insights into the photodissociation dynamics of small gas phase molecules that can be achieved using cutting-edge time resolved pump-probe spectroscopy methods (here using XUV absorption as the probe). Against this, I perceive two rival viewpoints. First, the opportunities afforded by this type of experiment have already been demonstrated, quite spectacularly in some cases, in other recent publications from the Berkeley group – which include papers describing studies of the simplest alkyl iodide (methyl iodide, ref. 36) and a somewhat richer variant, allyl iodide (ref. 40). Second, as the authors note in their introduction, the UV photochemistry of the alkyl (and related) iodides has already been much studied. What new insights are revealed in this manuscript? The key features, e.g. excited state potentials, relative transition dipole strengths, parent molecule excited state to atomic product state correlations, are all understood reasonably well and I/I* product branching ratios have been measured (often at several wavelengths) and rationalised in terms of branching at a conical intersection between the $3Q_0$ and $1Q_1$ potential energy surfaces – as summarised in the Introduction. Real time observation of the bifurcation at this conical intersection would be noteworthy but, as the authors point out (p. 3), ‘The determination of time constants characterising the evolution of the $3Q_0$ and $1Q_1$ features is currently limited by the temporal resolution of the experiments and is therefore not provided here’. Hence my dilemma: Those familiar with the field might well enjoy reading another well-written description of an application of an impressive new technology but will find few surprises in the results, while the more general chemical physics oriented reader could have already gained a similar overview from reading prior works from this group.

There is no doubt the science merits publication; my uncertainty is whether this particular work meets the stated criterion for publication in Nature Communications – namely that ‘it represents an advance in understanding likely to influence thinking in the field.’ But, given my expectation that the work will be published in some appropriate journal, I offer the following comments that the authors may wish to consider:

We thank the reviewer for providing feedback on our manuscript. In the following, we address the reviewer’s comments and questions.

As the reviewer mentioned, A-band dynamics in methyl iodide and allyl iodide investigated by means of XUV transient absorption spectroscopy have been previously reported. In those publications, short-lived transient spectral features assigned to molecular excited states were detected. However, the dynamics of the features were captured with limited temporal resolution. Consequently, the features could not be specifically assigned to either the 3Q_0 or 1Q_1 states and conical intersection dynamics could not be observed. In addition, spectral changes mapping continuous C-I bond elongation during photodissociation could not be fully captured. While the previous studies laid the conceptual groundwork for the experiments discussed in this manuscript, the present results provide the first direct observation of alkyl iodide dynamics in the A-band before and after the conical intersection that leads to control over photoproduct branching ratios, captures the continuous excited state evolution during

bond-breaking, and demonstrates the state-selectivity of the XUV technique in distinguishing 3Q_0 and 1Q_1 dynamics.

In the Introduction section of the manuscript on page 3, a sentence is added to emphasize that the present work represents the first direct observation of conical intersection dynamics in the alkyl iodide A-band:

Although a number of time-resolved experiments on the dynamics of alkyl iodide photodissociation in the A-band have been reported using femtosecond XUV transient absorption and Coulomb explosion imaging, limitations in the temporal resolution of previously-reported experiments have precluded a direct observation of passage through the conical intersection, which is a new achievement in this work.

- p. 2. The statement 'For few-carbon containing alkyl iodides, excitation to 3Q_0 comprises 70-80% of the oscillator strength of the A band' may need qualifying. For comparison with the present data, one is interested in the partial absorption cross-sections to the various excited states at the pump wavelength of interest. If transitions to excited states other than 3Q_0 in the present study really were responsible for 20-30% of the total oscillator strength, should they not be evident in the zero time delay transients?

If a decomposition of the A-band absorption is known, it is indeed possible to estimate the expected excitation probability to the 3Q_0 state based on the UV spectrum of the experimental pump pulse. On page 1 of the supplementary material, a discussion of excitation probabilities to the 3Q_0 state and their connection to the zero delay transients are added:

A-band absorption in the alkyl iodides consists of excitations to the optically accessible 3Q_0 , 1Q_1 , and 3Q_1 states. For *t*-C₄H₉I, a decomposition of the A-band absorption spectrum into relative absorption to the 3Q_0 , 1Q_1 , and 3Q_1 states has been obtained using magnetic circular dichroism. Based on the results of the decomposition and the measured spectrum of the UV pump pulse in the current experiment, excitation to 3Q_0 is expected to comprise ~85% of the total excitation of *t*-C₄H₉I. Meanwhile, weaker excitation to 1Q_1 and 3Q_1 is expected to comprise a minor ~3% and ~12% of the total excitation, respectively. The decomposition of the A-band absorption spectrum for *i*-C₃H₇I has not been reported in the literature. However, since the A-band decomposition is expected to be similar among the alkyl iodides such as CH₃I, C₂H₅I, and *t*-C₄H₉I, major excitation to 3Q_0 is also expected in the *i*-C₃H₇I experiment.

At 0 fs time delay in the experimental *i*-C₃H₇I and *t*-C₄H₉I transients (main text Fig. 2c,d), a strong 3Q_0 feature located at 47.1 eV dominates the spectrum. However, other features corresponding to minor excitations to the 1Q_1 and 3Q_1 states cannot be clearly identified. Since transitions to the 1Q_1 and 3Q_1 states are considerably weaker than transitions to the 3Q_0 state, the 1Q_1 and 3Q_1 states are likely to give signals at early time delays that are too small to be detected within the 2 mOD noise level of the experiments.

- Fig. 2. The authors might reflect on the colour scheme used to display in panels (a) and (b) (pale to dark with increasing time delay). This choice ensures that the dominant features are associated with the asymptotic I and I* atoms, yet the point of primary novelty is surely the variation in the transient spectra at short time delays – which is currently hard to make out. Might a multi-colour scheme allow clearer visualisation of the early time data?

In order to improve the visibility of the data, we have updated the colors used to plot the transient spectra to a multi-color scheme (Fig. 2a,b in manuscript page 4 and Supplementary Figs. 1,2 in supplementary material pages 3-4). The captions of the corresponding figures have been modified to read as follows: “The spectra are plotted in gray colors which evolve to blue/orange with increasing time delay.”

- p. 5 contains two curious sentences: ‘The modified CH₃I transients qualitatively reproduce with excellent agreement’; and ‘Future experiments with even shorter pump pulses will be the subject of a future study.’

We have rewritten the two sentences on page 5 and 6 in the manuscript for clarity as follows:

- In the modified CH₃I transients (Fig. 2e-f), changes in the spectral features in photon energy and intensity show strong similarities to those observed in the *i*-C₃H₇I and *t*-C₄H₉I results.
- Future experiments with shorter pump pulses, achieving faster temporal resolution, will allow time constants for the passage of *i*-C₃H₇I through the conical intersection to be more precisely characterized and compared to *t*-C₄H₉I, providing further insight into the influence of alkyl group structure on state-switching dynamics.

- p. 9. D.M.N. initials sequenced wrongly in line 3 of Author Contributions.

We thank the reviewer for noticing this typo. The initials are corrected to “D.M.N” on page 9 of the manuscript.

Response to Reviewer #2

This paper presents a very nice time-resolved study of the photodissociation of isopropyl- and t-butyl-iodide into an alkyl radical and an iodine atom in the I or I* spin-orbit state. Ultrafast transient absorption spectroscopy is used to observe passage through a conical intersection along the dissociation coordinate. The switch between potential surfaces leads to a switch in the absorption spectrum of the dissociating molecule as well as that of the ultimate fragments. What is great about the paper is that it is very clearly written, the effects are clearly visible and (mostly) readily understood by the reader, the theory presents some confirmation of the model, and the whole thing could be used as a terrific textbook example of the effects of conical intersections on photodissociation dynamics. The demonstration of the experimental approach, especially when combined with the theory, clearly suggests that it will have substantial long term value, and I am sure it will lead to lots of exciting avenues of research. I am not sure one could point to any one thing in the results on alkyl iodide photodissociation where you could say, these results demonstrate something that nobody knew before, but the example is so nice and the technique is so powerful that I believe that the paper should be accepted for publication in Nature Communications. As I said, the paper is very clearly written, and could be published as is. I just have a couple points the authors might want to consider.

We thank the reviewer for reading our manuscript and providing helpful suggestions. We also thank the reviewer for highly evaluating our work. Below, we address the reviewer's questions and suggestions.

1. On p. 3 the authors say, "the 3Q_0 state is observed primarily undergo strong transitions to the $(4d_{3/2})^{-1}\sigma^*$ state..., whereas the 1Q_1 state primarily undergoes strong transitions to the $(4d_{5/2})^{-1}\sigma^*$ state..." This makes it sound like the selectivity just happens to be there, but it seems like there must be a simple atomic picture that shows why this is true (and certainly the calculations reproduce it). Maybe they could put something in the supplementary material about this, and then change the text to make it sound like they know why it is that way?

As suggested by the reviewer, two specific changes to the text have been made.

- In the supplementary material on pages 7-8, we have added the following paragraph:

In this study, the 3Q_0 state primarily undergoes strong transitions to the $(4d_{3/2})^{-1}\sigma^*$ state, whereas the 1Q_1 state primarily undergoes strong transitions to the $(4d_{5/2})^{-1}\sigma^*$ state. The underlying cause of this selectivity in the strength of transitions from the molecular states is correlated with the strength of atomic iodine (I and I*) transitions, which are dictated by spin-orbit selection rules. Between the Franck-Condon regions of the 3Q_0 and 1Q_1 potentials and the dissociation limit, restrictions imparted by selection rules on the atomic transitions are partially but not fully lifted by interactions between the alkyl radical and iodine atom. Consequently, the selectivity of molecular transitions retains some of the selectivity of their corresponding atomic limits. Since atomic I* transitions to the $(4d_{5/2})^{-1}$ state are forbidden, 3Q_0 transitions to the $(4d_{5/2})^{-1}\sigma^*$ state are correspondingly weak. Similarly, since atomic I transitions to the $(4d_{3/2})^{-1}$ state are extremely weak, the strength of 1Q_1 transitions to the $(4d_{3/2})^{-1}\sigma^*$ state are likewise weak. Consequently, the molecular 3Q_0 state selectively undergoes strong transitions to the $(4d_{3/2})^{-1}\sigma^*$ state, whereas the molecular 1Q_1 state primarily undergoes strong transitions to the $(4d_{5/2})^{-1}\sigma^*$ state.

- In the manuscript on page 3, we have modified a sentence to read “In this study, the 3Q_0 state primarily undergoes strong transitions to the $(4d_{3/2})^{-1}\sigma^*$ state...whereas the 1Q_1 state primarily undergoes strong transitions to the $(4d_{5/2})^{-1}\sigma^*$ state due to spin-orbit selection rules imparted by the iodine atom.”

2. The pictures used in the paper show the motion along the reaction coordinate, but of course, even in the photodissociation of methyl iodide, the motion along other normal modes affects the branching and product distributions (e.g., the calculations of Guo and Schatz). The authors mention in the discussion that probing near the carbon edge could provide insight into those dynamics, which would be really neat. It would be interesting to understand the extent to which such motions responsible for the changing branching ratios in different alkyl iodides. Certainly, the translational energy distributions for both samples show broad distributions (see Refs. 29 and 30), reflecting significant excitation of the radical vibrational modes. In any case, in the presence of all these other degrees of nuclear motion, it is interesting that the passage through the conical intersection still happens so fast.

It is indeed interesting to note that the passage through the conical intersection occurs on a timescale of tens-of-femtoseconds, i.e. on the order of a typical vibrational period for normal modes of the molecule. For these systems, it seems likely that the rapidity at which the conical intersection passage proceeds is due to the location of the intersections and highly repulsive nature of the excited state potentials along the C-I bond. No changes to the text have been made.

3. I probably missed something in the explanation, but in Figure 2c, the population in $1Q_1$ evolves smoothly into the feature associated the free I atoms, but for the feature associated with $3Q_0$, there is a minimum of density in the evolution from the $3Q_0$ feature to the free I* atoms. This is reproduced in the simulation, but I am not quite sure why there is this minimum. Does it just reflect the very short period of time at the corresponding geometry? Why is it absent, or at least much less obvious, for the $1Q_1$ component?

These points are more curiosities, and I do think the paper can be accepted in the present form.

Indeed, in both the *i*-C₃H₇I experiment and the simulated CH₃I results there appears to be a transient minimum in signal density as the 3Q_0 feature evolves into the I* absorption line. Based on comparisons between the CH₃I simulations representing pure 3Q_0 evolution with and without time-broadening (shown below, panel A), the signal minimum observed in the CH₃I simulations appears to originate from an “inflection point” in the 3Q_0 feature where the shift in absorption energy is fastest (dotted circle). With time-broadening, blurring of the energetically shifting 3Q_0 feature leads to the prediction of an intensity drop in the signal. In the *i*-C₃H₇I experiment (Fig. 2c), we expect that this effect is at play. Currently, it is not clear if the inflection point reflects a very short period of time at the corresponding molecular geometry or if it is merely a consequence of the relative shapes of the 3Q_0 surface and core-excited surface, but it would make for a very interesting topic of a follow-up study.

As the reviewer mentioned, the 1Q_1 feature does not exhibit a significant drop in signal density. The absence is attributed to an absence of inflection points in the evolution of the 1Q_1 feature into the atomic I absorption line, as can be seen from the CH₃I simulations depicting the evolution of the 1Q_1 following conical intersection transfer (shown below, panel B).

In the supplementary material on page 11, a new figure (Supplementary Fig. 8) showing simulated CH_3I transients with increasing temporal broadening (from 5 to 50 fs) has been added. The figure allows the reader to more clearly observe how time-broadening of the shifting 3Q_0 spectral feature in the modified CH_3I transients leads to a transient intensity drop in the 3Q_0 signal.

Response to Reviewer #3

The group is one of the pioneer scientists to have developed the ATAS technique to reveal molecular dynamics. They have in the past studied several compounds using ATAS among which are some halogens :

- Direct Observation of the Transition-State Region in the Photodissociation of CH₃I by Femtosecond Extreme Ultraviolet Transient Absorption Spectroscopy ». JPCL 2015. (ref36)
- Transition State Region in the A-Band Photodissociation of Allyl Iodide—A Femtosecond Extreme Ultraviolet Transient Absorption Study ». JCP 2016 (ref40)
- Tracking Dissociation Dynamics of Strong-Field Ionized 1,2-Dibromoethane with Femtosecond XUV Transient Absorption Spectroscopy . PCCP(2016)
- Disentangling Conical Intersection and Coherent Molecular Dynamics in Methyl Bromide with Attosecond Transient Absorption Spectroscopy ». Nature Communications (2019) (ref15)

The group now presents a study of the iso-propyl iodide and tert-butyl iodide molecules (i-C₃H₇I and t-C₄H₉I) dissociation which is characterized by a larger amount of I/I* branching ratio relative to smaller alkyl iodides. The introduction, experimental results and their assignments are perfectly written and explained with the proper references. The reading is very easy and the experimental data beautiful to see (only 2mOD ! Thanks to the high-frequency post-process analysis). It was a real pleasure to read it. If the below mentioned concerns and minor items can be addressed satisfactorily, one could consider publication in Nat. Com.

We thank the reviewer for the helpful suggestions and for the praise of the work. Below, we address the reviewer's comments and questions.

Main concerns :

1) In Fig2c for i-C₃H₇I, why does the feature at 47.6 eV, that is fragment signature , seems dominant before 120fs and not after ?

In the *i*-C₃H₇I experiment (Fig. 2c), the Δ OD of the atomic I fragment signal at 47.6 eV increases to 4 mOD between 0 and 120 fs and decreases slightly from 4 mOD to 2 mOD between 120 and 160 fs. We also note that in the atomic I fragment signal at 45.9 eV, the Δ OD increases to 30 mOD between 0 and 120 fs and slightly decreases from 30 mOD to 28 mOD between 120 and 160 fs. In both cases, the change in Δ OD is 2 mOD, which is within the noise level of the current experiment. Consequently, it is difficult to determine at this time whether the measured decreases in signal are real or due to experimental noise. In the supplementary materials on page 5, a figure showing normalized time traces taken along the atomic I fragment signal at 45.9 eV is shown in Supplementary Fig. 3c. The figure allows the reader to more clearly observe transient changes in the spectrum at 45.9 eV.

2) In fig2d for t-C₄H₉I, what is the assignment of transients at 46.4-46.8 eV/40-90 fs?

In the *t*-C₄H₉I transient (Fig. 2b,d), the broad, featureless signal observed between 46.4-46.8 eV / 40-90 fs results from a region of signal overlap between adjacent I(45.9 eV) and ³Q₀(47.1 eV) peaks. The observation that the broad overlap signal somewhat rises and decays between 40-90 fs is attributed to the rise in I(45.9 eV) and decay of ³Q₀(47.1 eV), changing the signal level in the overlap region. On page 4

of the manuscript, a sentence has been added to the Fig. 2c,d caption mentioning the origin of the signal: “In the 46.4-46.8 eV region, adjacent I(45.9 eV) and ³Q₀(47.1 eV) peaks can overlap to create a background signal.”

3) It is really disappointing that there is no attempt to fit the rising and decay times of the features by a deconvolution with the apparatus function. Indeed, when the precision of the cross-correlation is +/-7fs, it means that the resolution of the dynamics will appear on the same range. It is especially true to go beyond the contour plot that gives the misleading first impression that product features at 45.9 eV appears as soon as 0fs for *i*-C₃H₇I. Similar treatments were already made in the JCP 2016 for allyl iodine with a longer cross-correlation. If it is due to the limited space of Nat.Com, it could be included in the SI. It is also relevant to compare the two molecular systems and to conclude if the time-scales can also rationalize the different I*/I ratio observed.

We agree with the reviewer’s assessment that further information on rise and decay dynamics can be included. On pages 4-6 of the supplementary material, a discussion of time traces taken along features of the experimental transients is added to the text and presented as follows:

In the experimental *i*-C₃H₇I and *t*-C₄H₉I transients, XUV features at 47.1, 45.4, and 45.9 eV appear in the spectrum. Time traces taken along these features are shown in **Supplementary Fig. 3**. The dynamics of the features are modeled by exponential functions convolved with the Gaussian instrument response. The time trace at 47.1 eV (**Supplementary Fig. 3a**) is fit to an exponential decay:

$$S_{\text{decay}}(\tau) = \text{Heaviside}(\tau-t_0) * A \exp(-(\tau-t_0) / \tau') \otimes \exp(-4 \ln(2) (t/\tau_{\text{IRF}})^2) \quad (\text{Eq. 2})$$

where τ corresponds to the experimental time delay, τ_{IRF} is the measured instrument response time (50 fs), and the proportionality constant (A) and the time constants (t_0 and τ') are free parameters in the fitting. The time trace at 45.9 eV (**Supplementary Fig. 3c**) is fit to an exponential rise:

$$S_{\text{rise}}(\tau) = \text{Heaviside}(\tau-t_0) * A (1-\exp(-(\tau-t_0) / \tau')) \otimes \exp(-4 \ln(2) (t/\tau_{\text{IRF}})^2) \quad (\text{Eq. 3})$$

where the proportionality constant (A) and the time constants (t_0 and τ') are free parameters in the fitting. The 45.4 eV time traces (**Supplementary Fig. 3b**) are heavily convolved with the instrument response function in the *i*-C₃H₇I and *t*-C₄H₉I experiments, and are challenging to model due to the many-parameter fitting required to capture the exponential rise and decay of the time trace, precluding an accurate fitting. Time constants obtained from fitting the time traces at 47.1 and 45.9 eV are summarized in **Supplementary Table 1**.

The timescales of passage through the conical intersection can be estimated from a time trace taken along the decaying signal at 47.1 eV. The time traces are found to decay with a time constant of $\tau' = 36 \pm 4$ fs for *t*-C₄H₉I and $\tau' = 67 \pm 6$ fs for *i*-C₃H₇I. For molecules in which nearly complete state-switching at the conical intersection occurs, the decay of the signal at 47.1 eV corresponds primarily to ³Q₀ (*Region 1*, main text Fig. 1a) population decay via state-switching at the conical intersection with a time constant of $\tau_{\text{CI}} \approx \tau'$. Passage through the conical intersection in *t*-C₄H₉I is therefore estimated to occur with a time constant of $\tau_{\text{CI}} \approx 36 \pm 4$ fs. In the case of partial state-switching, transient features corresponding to ³Q₀ (*Regions 2-3*) may spectrally overlap the signal at 47.1 eV, resulting in a measured decay constant τ' that is slightly longer than the time constant associated with state-switching. Consequently, state-switching dynamics in *i*-C₃H₇I at the conical intersection are estimated to occur with a time constant $\tau_{\text{CI}} \leq 67 \pm 6$ fs.

The time trace at 45.9 eV is associated with wave packet motion into the asymptotic region of the 1Q_1 surface and the atomic I dissociation limit (*Region 3*). The exponential rise time of the 45.9 eV time trace is found to be slower in *t*-C₄H₉I compared to *i*-C₃H₇I, suggesting that C-I bond elongation in the asymptotic region of the 1Q_1 surface proceeds more slowly in *t*-C₄H₉I. This observation is consistent with previous femtosecond velocity-map imaging experiments in which the rise times for atomic I photodissociation products were found to be faster in *i*-C₃H₇I compared to *t*-C₄H₉I.

Supplementary Fig. 3. Normalized time traces obtained from the experimental *i*-C₃H₇I transient (blue) and *t*-C₄H₉I transient (orange). The time trace data (dashes/circles) are fit to exponential functions convolved with the Gaussian instrument response (dotted gray curves). The results of the fittings are plotted as solid curves. **a** Time trace along 47.1 ± 0.1 eV (dashes), assigned to transitions from 3Q_0 before the conical intersection and fit to Eq. 2 (solid curves). **b** Time trace along 45.4 ± 0.1 eV (dashes), assigned to transitions from 1Q_1 following the conical intersection. **c** Time trace along 45.9 ± 0.1 eV (circles), assigned to transitions from the asymptotic region of 1Q_1 into the atomic I dissociation limit and fit to Eq. 3 (solid curves).

Species	Position of time trace (eV)	t_0 (fs)	τ' (fs)
i -C ₃ H ₇ I	47.1	-1 ± 1	67 ± 6
	45.9	17 ± 5	26 ± 8
t -C ₄ H ₉ I	47.1	0 ± 2	36 ± 4
	45.9	19 ± 4	42 ± 7

Supplementary Table 1. Time constants for *i*-C₃H₇I and *t*-C₄H₉I obtained from fitting the time traces at 47.1 eV and 45.9 eV to Eqs. 2 and 3, respectively. The reported uncertainties in the time constants correspond to the 95% confidence intervals obtained from the fittings.

On page 5 of the manuscript, the sub-100 fs decay of the 3Q_0 feature is now further quantified:

Based on an analysis of time traces taken at 47.1 eV in the molecular transients (see Supplementary Note 3), 3Q_0 population is found to decay through state-switching at the conical intersection with a $\tau_{CI} = 36 \pm 4$ fs exponential time constant in *t*-C₄H₉I and $\tau_{CI} \leq 67 \pm 6$ fs exponential time constant in *i*-C₃H₇I. Although exponential time constants associated with state-switching at the conical intersection can also be obtained from time traces at 45.4 eV in the molecular transients, the analysis is currently limited by the temporal resolution of the experiments.

4) I don't understand what is clearly done in 'enforcing the convergence of I:I* branching ratios to values of 2:1 and 13:1 at 50 fs time delay'. A few sentences should be written to tell the reader that the dynamics in the simulation is twice as fast as in the experiment. Could this simulation be temporally broadened by a 50 fs Gaussian (as in the experiment) instead of 15 fs which does not make sense to me.

- On page 10 of the supplementary material, the sentence is updated for improved clarity and now reads as follows:

Direct sums between the two representative XUV transients in which the I:I* ratios are taken to be 2:1 and 13:1 were used to produce the modified transients shown in Fig. 2e-f of the main text.

- As suggested by the reviewer, on page 10 of the supplementary material we have added a few sentences to emphasize that the dynamics in the simulations for CH₃I are faster than those in the *i*-C₃H₇I and *t*-C₄H₉I experiments:

Conical intersection and dissociation dynamics in the simulated CH₃I transients appear to evolve on a faster timescale than in the *i*-C₃H₇I and *t*-C₄H₉I transients. The appearance of faster dissociation dynamics in the simulated CH₃I transients is consistent with the expectation that separation of the C-I bond in CH₃I may proceed more quickly as compared to the heavier *i*-C₃H₇I and *t*-C₄H₉I systems.

- In the supplementary material on page 10, a few sentences have been added clarifying how the selection of a 15 fs temporal broadening was made:

The calculated CH₃I transients presented in Fig. 2e-f of the main text are temporally broadened by a Gaussian blur to facilitate qualitative, visual comparisons to the experimental *i*-C₃H₇I and *t*-C₄H₉I transients. As shown in **Supplementary Fig. 8**, the modified CH₃I transients with 15 fs Gaussian broadening allows for the closest reproduction of features observed in the *i*-C₃H₇I and *t*-C₄H₉I transients (Fig. 2c-d, main text).

Supplementary Fig. 8. Modified CH₃I transients with 5 fs, 15 fs, or 50 fs Gaussian broadening applied. a-c Modified CH₃I transients converging to the branching ratio I:I* = 2:1. d-f Modified CH₃I transients converging to the branching ratio I:I* = 13:1. The color scales of the transients are scaled independently.

In addition, Supplementary Fig. 7 on page 10 of the supplementary material is modified to include a label "15-fs blur" to emphasize that a 15 fs broadening is applied to the displayed simulations.

Minor ones

5) I would have expected a slight energy variation of the 1 and 2 features due to the chemical shift variation with the C-I distance. This is not observed: is it due to the long 50 fs duration of the pump pulse? It would be great to discuss the requirement on the experimental set-up to observe on such compounds a chemical shift variation.

As the reviewer suggested, the magnitude of the energy shifts in the experiment can be limited by temporal broadening by the pump pulse. In general, shifts in transition energy can be observed as long as the magnitude of the shift is greater than 40 meV (the experimental energy resolution) and evolves on a timescale slower than 50 fs (the experimental time resolution). In the supplementary material on page 6, the energy shifts in the 3Q_0 feature are analyzed:

Changes in the position of the 3Q_0 feature in the *i*-C₃H₇I and *t*-C₄H₉I spectra are indicative of wave packet motion on the 3Q_0 surface. For both molecules, changes in the position of the 3Q_0 feature are tracked by fitting the feature to a Gaussian function at each time delay point, and extracting the center energy position of the Gaussian fit. The obtained 3Q_0 peak positions are plotted as a function of time in **Supplementary Fig. 4**. For *t*-C₄H₉I, the extracted positions of the 3Q_0 feature are truncated at 70 fs time delay, after which the signal level is too low to allow for a Gaussian fitting. In the *t*-C₄H₉I experiment, the 3Q_0 peak position remains relatively constant at 47.1 eV within the current experimental energy resolution (40 meV). In contrast, the 3Q_0 peak position in the *i*-C₃H₇I experiment is observed to shift from an initial position of 47.1 eV into the I*(46.7 eV) absorption line at long time delays, concomitant with dissociative wave packet motion on the 3Q_0 surface.

Supplementary Fig. 4. Position of the 3Q_0 feature in the *i*-C₃H₇I transient (blue curve) and *t*-C₄H₉I transient (orange curve) as function of time delay. The evolving photon energy position (solid curve) of the 3Q_0 feature is tracked by a Gaussian fitting at each time delay point of the transient spectra. The 95% confidence intervals in the position of the 3Q_0 feature obtained from the Gaussian fits are plotted in blue and orange shades. The peak position of the atomic I*(46.7 eV) transition is marked with a dotted line.

6) I don't like the figure 1A because it gives the wrong impression that R products appear without internal energy and that all the released energy is of kinetic nature relative to the center of mass.

We thank the reviewer for pointing out that Fig. 1a does not directly express the possibility of internal energy in the R group. Therefore, we have revised the caption of Fig. 1a on page 2 of the manuscript to include the following clarifying sentence: "The partitioning of available energy into different degrees of freedom is not represented in this schematic."

REVIEWERS' COMMENTS:

Reviewer #3 (Remarks to the Author):

All my comments have been considered in the revised version, the paper can be published as it is.

NCOMMS-20-16239 B - Reviewer Responses

Contained below are responses to Reviewer #3, following the second manuscript submission:
NCOMMS-20-16239A.

Normal text: reviewer comments
Orange text: our response

Response to Reviewer #3

All my comments have been considered in the revised version, the paper can be published as it is.

We thank the reviewer for their comments on the manuscript and the evaluation of its suitability for publication.